# α_1A_-Adrenergic Receptor as a Target for Neurocognition: Cautionary Tale from Nicergoline and Quinazoline Non-Selective Blockers

**DOI:** 10.3390/ph18101425

**Published:** 2025-09-23

**Authors:** Dianne M. Perez

**Affiliations:** Department of Cardiovascular & Metabolic Sciences, Lerner Research Institute, The Cleveland Clinic Foundation, 9500 Euclid Ave, Cleveland, OH 44195, USA; perezd@ccf.org; Tel.: +1-216-444-2058

**Keywords:** Alzheimer’s disease, adrenergic receptor, GPCR, α-AR blockers

## Abstract

Decades ago, previous studies that used non-selective ergot derivatives suggested that blockage of the α_1A_-adrenergic receptor mildly increased cognition through increased blood flow to the brain due to vasodilation and, thus, could be used as a treatment for dementia. However, further studies indicated that nicergoline was non-specific and hit many different targets. Today, a similar scenario is developing with the use of non-selective α_1_-AR antagonists of the quinazoline class, referred to as “osins”, as potential treatments for COVID-19/SARS, post-traumatic stress disorder, cancer, and neurodegenerative disorders, such as Parkinson’s, Alzheimer’s, and amyotrophic lateral sclerosis. While there is extensive evidence of neuroprotection from many clinical trials, the mechanism of action of quinazolines is often not α_1_-AR-mediated but keyed to its glycolysis-enhancing effects through activation of the enzyme phosphoglycerate kinase 1 (PGK1). These studies have incorrectly labeled the α_1A_-adrenergic receptor as an “old target” to treat Alzheimer’s and other neurocognitive diseases, hampering drug development. This review will summarize these and other studies to indicate that activation, not blockage, of norepinephrine’s actions, through α_1A_-AR, mediates cognitive, memory, and neuroprotective functions that may reverse the progression of neurocognitive diseases.

## 1. Introduction

### Norepinephrine and the Adrenergic Receptors: Each Subtype Can Have Distinct Functions

Adrenergic Receptors (ARs) are G-Protein Coupled Receptors (GPCRs) that regulate neurotransmission and the sympathetic nervous system through different ARs that bind the neurotransmitter, norepinephrine (NE), and the neurohormone, epinephrine (EPI). Drugs that target GPCRs represent about 30% of all current clinical drugs because of the receptor’s cell surface localization and well-characterized pharmacological and physiological hormonal functions. There is a total of nine AR family members (β_1_, β_2_, β_3_, α_2A_, α_2B_, α_2C_, α_1A_, α_1B_, and α_1D_) that bind NE/EPI, but they can regulate distinct functions through coupling to different G-proteins and signaling pathways. Many of the distinct functions of each subclass of ARs are driven by tissue localization, relative density, and signal transduction differences.

β-ARs couple mainly through Gs, a GTP-binding protein (G-protein) that stimulates adenylate cyclase, to produce cyclic adenosine 3′, 5′,-monophosphate (cAMP), which then activates the protein kinase A (PKA) signaling pathway. Because it was the first GPCR that was cloned, subtype-distinct functions are more defined, facilitating drug development. β_1_-ARs are highly expressed in the heart. β-blockers are used to slow down the heart rate and reduce cardiac workload to treat heart failure. β_2_-ARs are highly expressed in the lungs. β_2_-ARs agonists are used to treat asthma by dilating the smooth muscle of the bronchioles to increase air flow to the lungs. β_3_-AR agonists are being explored to regulate metabolic disorders and overactive bladder [1].

α_2_-ARs couple mainly through Gi, a GTP-binding protein that inhibits adenylate cyclase, which decreases the production of cAMP, and are often used to regulate the cAMP levels induced through β-ARs. A well-known regulation of this type is insulin secretion. α_2A_- and α_2C_-ARs are known to regulate neurotransmission by their location on synaptic terminals. Actions of α_2A/C_-AR inhibit the release of NE, which is referred to as an auto-receptor, but do so at different sympathetic stimulation frequencies [2]. α_2A_-AR activation also leads to a decrease in blood pressure, while α_2B_-AR stimulation may counteract this effect by causing direct vasoconstriction. α_2C_-ARs participate in vasoconstriction after exposure to cold temperature [3]. Clinically, several types of non-selective α_2_-AR agonists, such as clonidine, medetomidine, and brimonidine, are being used to treat patients for sedation or for a variety of symptoms, such as hypertension, glaucoma, tumor pain, postoperative pain, shivering, or to block the symptoms of sympathetic overactivity, but all produce unwanted side effects.

α_1_-ARs were the last of the ARs to be cloned and characterized. α_1_-ARs canonically couple to Gq, a GTP-binding protein that activates phospholipase C (PLC), which causes the hydrolysis of membrane-bound phosphatidylinositol 4,5-bisphosphate to release inositol triphosphate (IP3) and diacylglycerol (DAG). IP3 binds to receptors located on the endoplasmic reticulum, causing the release of calcium that constricts the smooth muscle in blood vessels, increasing blood pressure. DAG activates protein kinase C (PKC), an enzymatic effector that phosphorylates many proteins to amplify signals downstream in the signaling cascade. The non-selective α_1_-AR antagonists, prazosin, terazosin, doxazosin, tamsulosin, and alfuzosin are approved to treat the symptoms of benign prostatic hyperplasia by relaxing prostatic smooth muscle to improve urinary flow. They are taken at night to minimize effects on blood pressure. Because the receptor target invokes unwanted side effects on blood pressure, subtype-specific drug development for the clinic has not been a high priority. However, transgenic and knockout (KO) mouse models have identified some key subtype-selective functions that may be targeted for subtype-selective drug development. It is now recognized that α_1A_-AR activation is cardioprotective in heart failure and ischemia, while α_1B_-AR overactivity is cardiac maladaptive [4]. As detailed later, α_1A_-AR activation has also been shown to be cognitive and memory enhancing, while α_1B_-AR activation is pro-epileptic and neurodegenerative [5]. α_1_-ARs, as with all GPCRs, can signal directly through their main signaling cascade or through crosstalk to increase signal diversity via G protein-dependent and independent pathways, spatio-temporal mechanisms, and biased agonistic signaling [6,7,8,9].

There is evidence that each AR family and subtype can regulate distinct functions and physiology and has the potential to be therapeutically targeted to alleviate disease symptoms. However, there is a distinct lack of sufficiently selective activators or blockers that has hampered these assessments, and the use of non-selective agents leads to many unwanted side effects and may have additional off-targets; such is the case for nicergoline and the non-selective quinazoline α_1_-AR antagonists.

This review will summarize the pharmacological and therapeutic functions of nicergoline and the non-selective quinazoline α_1_-AR antagonists, prazosin, doxazosin, terazosin, and alfuzosin (Figure 1) as they pertain to neuroprotection and remediation of the symptoms of dementia or Alzheimer’s disease. Decades of research have supported the hypothesis that activation, not blockage, of NE function through its many subtypes of ARs is pro-cognitive and may be of therapeutic benefit in Alzheimer’s disease. In the case of nicergoline, this review will highlight the numerous physiological functions and complex mechanism of action mediated through targeting several neurotransmitters, not just α_1_-AR. In the case of non-selective quinazoline α_1_-AR antagonists, which are prescribed to treat benign prostatic hyperplasia (BPH) through the relaxation of smooth muscle, this review will highlight their ability to activate PGK1, a non-α_1_-AR-mediated mechanism, to increase glycolysis to remediate neurodegenerative diseases. Studies that explore the cognitive ability of tamsulosin, an α_1_-AR antagonist also used to treat BPH, but does not contain the quinazoline pharmacophore and is somewhat selective for the α_1A_-AR subtype, have shown that it does not bind and activate PGK1 and displays opposite cognitive functions compared to quinazoline α_1_-AR antagonists. Thus, this supports the interpretation that quinazoline α_1_-AR antagonists’ effects, as they pertain to neurodegeneration and Alzheimer’s disease, are non-α_1_-AR-mediated and “off-target”. In both scenarios of blocking α_1_-ARs by either nicergoline or quinazoline α_1_-AR antagonists, the interpretation that inhibiting α_1_-AR activation would be of therapeutic benefit in dementia or Alzheimer’s disease is flawed.

## 2. NE and AR Activation Increase Memory and Cognition

There is an abundance of evidence that NE activation enhances learning, memory, and neuroprotective signals [10,11]. Cognitive enhancement can be achieved through direct stimulation of the ARs or through the use of selective NE reuptake inhibitors to increase NE levels in the frontal cortex or hippocampus [12]. NE is the main neurotransmitter synthesized by locus coeruleus neurons that degenerate and is an early pathology in Alzheimer’s disease [13]. Mice unable to synthesize NE by genetic deletion of the dopamine β-hydroxylase gene show cognitive dysfunction, synaptic, and long-term potentiation (LTP) deficits [14,15]. Specific areas of learning and memory affected by NE include focused and flexible attention, increased arousal and alertness, working memory, and memory formation and retrieval [5]. Using KO and overexpressed mouse models, in addition to various clinical studies, the activation of all AR subtypes (β, α_2_, α_1_) indicated some level of enhanced cognitive function [16,17,18]. Of note, meta-analysis of clinical trials where NE was stimulated with various agonists of the AR family has established a beneficial effect on cognition [19]. The memory/cognitive-inducing signals of NE primarily involve cyclic adenosine monophosphate (cAMP) production, phosphorylation of cyclic AMP response element-binding protein (CREB), or Exchange Proteins Activated by cAMP (EPAC) [20,21,22], which α_1_-ARs and β-AR canonical pathways can also regulate [23,24,25,26]. In addition, Extracellular Signal-Regulated Kinase (ERK) is also an NE-mediated cognitive and neuroprotective signal [20,27,28], also mediating the transcription and translation of proteins that increase metaplasticity [29].

## 3. α_1_-AR Activation Increases Cognition and Memory

α_1_-AR activation, particularly the α_1A_-AR subtype, has significant roles in the regulation of synaptic efficacy, both short- and long-term synaptic plasticity, and different types of memory [5]. Most brain areas affected in Alzheimer’s disease express α_1_-AR, and they mediate various functions of learning and memory. α_1_-ARs increased LTP and LTD (long-term depression) in the prefrontal cortex, neocortex, ventral tegmental area, and hippocampus, and are associated with increased cognition. Both LTP and LTD may impart different forms of synaptic information during spatial learning. α_1_-AR activation can enhance memory recall, retention, and consolidation in the entorhinal cortex, fear-conditioned memory in the amygdala, and spatial memory and associative learning in the prefrontal cortex and hippocampus [5].

The generation of transgenic mouse models of the α_1_-AR subtypes and the development of highly selective subtype-selective ligands that can discriminate the subtypes have supported the hypothesis that activation of NE through α_1A_-AR generates a pro-cognitive and neuroprotective profile through regulation of neurogenesis and both short-term and long-term synaptic plasticity [5]. While all three of the α_1_-AR subtypes are associated with various aspects of learning and memory, α_1A_-AR shows the most promise as a therapeutic in AD. α_1A_-AR protein and RNA levels are downregulated in AD and in an AD mouse model [30,31], and polymorphisms are linked with AD [32] and schizophrenia [33]. In addition, a positive allosteric modulator of α_1A_-AR was shown in pre-clinical studies to improve long-term synaptic plasticity and cognition, and clear β-amyloids in AD mouse models better than donepezil (i.e., Aricept) [34].

## 4. Nicergoline, an Ergot Derivative Originally Proposed to Treat Dementia Through Vasodilation, Does Not Specifically Block α_1A_-AR 

While there is substantial evidence supporting the activation of α_1A_-AR as a therapeutic route to treat AD, some previous studies have suggested that blocking α_1_-AR through nicergoline would result in a pro-cognitive profile due to an increased blood flow to the brain. As reviewed below, while nicergoline has a pro-cognitive profile, it is not mediated through α_1A_-ARs; however, through the years, this outdated understanding of nicergoline has muddled the field of α_1_-AR-based therapeutic development.

Nicergoline (Figure 1), chemically defined as 8-beta-(5-bromonicotinoylhydroxymethyl)-1,6-dimethyl-10alpha-metoxyergoline, is a bioactive, alkaloid molecule derived from ergot fungus that was first described more than five decades ago as a neuroprotective agent to treat dementia-related conditions in the elderly. Nicergoline is known under various trade names such as Sermion and Adavin, but is not FDA-approved. In fact, nicergoline was banned in Europe in 2013 along with other ergot derivatives because of its side effects. However, its noted efficacy in clinical trials to improve cognition, but incorrect initial assessment on selective α_1A_-AR antagonism has led to the speculation that blocking, not activating this receptor, would be therapeutic for neurocognitive diseases.

Most neurological studies using nicergoline focused on cerebrovascular disorders, and there is only limited data for AD. Since 1972, hundreds of clinical trials using nicergoline have been performed with various criteria and endpoint evaluations. In 2001, a meta-analysis of 11 of these clinical trials was performed that included only studies that were double-blinded and placebo-controlled and assessed mostly older patients with cognitive impairment from a wide variety of clinical origins. The results of this meta-analysis provided evidence that nicergoline can improve cognition but also indicated some problems with the tolerance of this medication [35]. Since then, more recent clinical studies have confirmed that nicergoline enhances the cognitive performance of patients with dementia [36]. Nicergoline also confers neuroprotective benefits, which aid in potentially lessening age-related cognitive decline [37,38]. However, the drug has never been compared to other therapies, such as donepezil, that treat cognitive disorders as a benchmark. Paradoxically, a recent meta-analysis of 31,881 dementia-related reported adverse events has identified that nicergoline has a high risk of inducing the dementia it was thought to protect against [39].

Cognitive benefits of nicergoline may be through increased vasodilation, resulting in increased cerebral and peripheral blood flow, improving the symptoms of vascular dementia and neurodegeneration, but the effects are complex due to simultaneously blocking α_1_-ARs, centrally acting α_2_-ARs, and other neurotransmitter receptors [40]. In fact, various studies (dog, cat, rabbit, rat, mouse, and guinea pig) showed that nicergoline affects blood pressure and heart rate only slightly and increases the blood flow in the brain and hind limb without affecting the splanchnic and aortic flow in normal animals, indicating that not only α_1_-ARs are being blocked [41,42]. Nicergoline has numerous pharmacological and physiological effects in addition to vasodilation, such as increased cholinergic and catecholaminergic activity, increased metabolism, antioxidant and neurotrophic effects, and anti-platelet aggregation, most likely due to its affinity for many different receptor systems [43]. Cognitive benefits can also be due to increased acetylcholine release [44] and enhancement of choline acetyltransferase activity [45], which correlated with improvement in rodent memory tests [46]. In addition, nicergoline is neurotrophic, increases nerve growth factor in aged rats [47], and induces antioxidant effects [48,49] that may protect against the loss of cholinergic neurons.

The very broad spectrum of nicergoline’s functions is consistent with its non-selective nature. Originally described as a vasodilating α_1_-AR blocker [50] and later as an α_1A_-AR selective antagonist (pA2 = 8.8 or 3 nM) [51], this classification was solely based upon its sensitivity to the alkylating agent, chloroethylclonidine (CEC), which has been shown to be non-selective [52,53] and not useful in discriminating between the α_1_-AR subtypes. In addition to blocking α_1_-AR non-selectively, nicergoline has similar affinity in blocking several serotonin receptors (IC_50_ = 6 nM) and shows moderate affinity for dopamine, α_2_-ARs, and muscarinic acetylcholine M_1_ and M_2_ receptors [37,38]. Unfortunately, most of these studies used tissue preparations and were not pharmacologically characterized in any cloned and isolated receptor systems, which were available and are considered a gold standard to characterize a ligand’s selectivity. The IC_50_ of nicergoline in vitro has been reported to be 0.2 nM [54], about 10-fold higher affinity than any tested receptor, suggesting there may be additional unknown targets. Considering the non-selective nature of nicergoline, the previous hypothesis that cognition increases due to blocking α_1A_-AR activity does not seem likely.

## 5. Non-Selective Quinazoline-Derived α_1_-AR Blockers—The “Osins” Cause “Off-Target” Neuroprotective Effects

While nicergoline does not share any obvious structural features with common α_1_-AR blockers (Figure 1), the broad neuroprotective effects of the non-selective α_1_-AR antagonists, prazosin, doxazosin, and terazosin (often referred to as the “osins”) are well-documented [54,55,56,57,58,59,60,61,62,63]. These drugs are approved and considered a first-line treatment for BPH by the relaxation of prostatic and bladder smooth muscle, a basic α_1_-AR physiological function [64]. However, neuroprotective effects are not α_1_-AR-mediated but through the binding and activation of phosphoglycerate kinase (PGK1)-mediated ATP production. PGK1 is the first enzyme in the glycolysis pathway and converts ADP into ATP, which can fuel the high-energy requirements of the brain. Terazosin increases the release of ATP by competing for the same binding site as ADP in PGK1, thereby exerting an agonistic effect [65]. However, at high concentrations, it can inhibit PGK1 activity [66].

Neuroprotection may be mediated through metabolism-based therapies. In a degenerating motor neuron-based model of amyotrophic lateral sclerosis, increased glucose uptake and metabolism are neuroprotective [67]. As heart failure is considered an energy-starved disease [68], so are several neurodegenerative diseases. The brain utilizes glucose as its primary fuel for its high energy demands, accounting for 20% of whole-body energy consumption, but comprises only 2% of body mass [69,70]. In congruence, bioenergetic and mitochondrial dysfunction are commonly seen in neurodegenerative diseases and can modulate onset and progression [67,71,72]. It is hypothesized that increased PGK1-mediated ATP availability in neurons allows better adaptation to the cellular challenges of aging and protein aggregation, supported by epidemiologic cohort studies of the “osins” in the treatment of Parkinson’s disease [73,74] and among patients using “osins” for benign prostatic hyperplasia (BPH) [75].

While the neuroprotective benefits of quinazoline-based α_1_-AR antagonists are substantially evidenced, there is conflicting evidence on their role in mediating cognition as a primary outcome. A clinical trial in AD patients concluded that prazosin was effective in relieving agitation and aggression, but cognition was not assessed [76]. A meta-analysis of seven independent clinical studies found no clear association between non-selective α_1_-AR antagonists and modulation of cognition, with varied results indicating increased, decreased, or no change in the risk of developing dementia [77]. However, all of the neuroprotection and/or cognitive studies utilized these non-selective antagonists for the α_1_-AR subtypes and not any other α_1_-AR antagonists that are more subtype-selective, such as 5-methylurapidil. In general, any agent that reduces blood pressure can have indirect effects on cognition in the elderly and delay the risk of developing dementia [78], but there is a lack of direct evidence that non-selective quinazoline antagonists of α_1_-ARs can alter cognition and whether it is through PGK1 or α_1_-AR activity.

Quinazolines are an interesting chemical moiety and have garnered considerable interest in drug development. It was originally discovered in febrifugine, a quinazoline alkaloid with antimalarial potential [79]. Pharmacologically active molecules based on quinazoline scaffolds are strong chemotherapeutic drugs with anticancer, antimicrobial, antioxidant, anti-inflammatory, and antidiabetic properties [80]. As discussed previously, quinazolines also activate PGK1. Interestingly, quinazolines share their pharmacophore with that of acetylcholinesterase inhibitors (AChEIs), which are used as a current treatment for Alzheimer’s disease. A field-based 3D-QSAR (Quantitative Structure–Activity Relationship) pharmacophore design model of quinazoline-based AChEIs identified three novel lead molecules as potent AChEIs [81]. Quinazoline can also inhibit monoamine oxidases (MAO); the MAO-A subtype metabolizes NE, epinephrine, and serotonin. An inhibitor would increase the levels of these neurotransmitters and is also used to treat Alzheimer’s disease and Parkinson’s [82]. As Alzheimer’s disease is complex and involves multiple pathways, a drug possessing a single target mechanism may not be effective enough. Quinazolines hold promise in this regard.

## 6. Tamsulosin Is Not a Quinazoline and May Increase Risk for Dementia

PKG1 binding and activation have also been demonstrated in other related quinazoline-related α_1_-AR antagonists such as alfuzosin, prazosin, and doxazosin (Figure 1) [65], but not tamsulosin, an α_1_-AR blocker with some selectivity (10-fold) for α_1A/D_-AR [83,84]. Tamsulosin does not contain the quinazoline motif and does not interact with PGK1 [55], supporting that quinazoline α_1_-AR antagonists have “off-target” effects. Tamsulosin shares some structural motifs with another somewhat selective non-quinazoline α_1A_-AR antagonist, WB4101 (Figure 1). Tamsulosin also does not appear to mediate anti-inflammatory or neuroprotective effects [63,76], thus confirming that quinazoline’s neuroprotective effects are non-α_1_-AR-mediated. In further support, a cohort study of individuals treated with terazosin, alfuzosin, or doxazosin for urinary problems indicated a lower risk of developing Parkinson’s disease (PD) when compared to patients treated with tamsulosin [73]. In fact, tamsulosin was suggested to even potentiate PD progression [75], increase the risk of dementia in older men with BPH [85], and impair memory acquisition and consolidation in mice [86], suggesting that blocking α_1A_-AR impairs cognition. While designed to be α_1A_-AR selective, tamsulosin has 10-fold selectivity only against the α_1B_-AR subtype but has equal affinity between α_1A_- and α_1D_-AR [83,84]. In contrast to the potential anti-cognitive effects of tamsulosin and recent work indicating that a highly selective α_1A_-AR positive allosteric activator can reverse AD in pre-clinical studies in two different AD mouse models [34], a recent publication using a neuron-specific intracerebroventricular adeno-associated viral delivery of short-hairpin RNA to knockdown the α_1A_-AR in a 3xTG AD mouse model indicated reversal of cognitive, neuroinflammatory deficits and taupathology [87]. However, receptor expression levels were not assessed using a radiolabeled binding study, but were assessed using a peptide-generated polyclonal antibody, which was not verified against α_1A_-AR knockout (KO) tissues. These antibodies have been tested in two independent studies using the KO mouse models as negative controls and indicated that all commercially available antibodies against the α_1_-AR subtypes are non-specific [88,89]. In addition, confirmatory studies in [87] were performed using terazosin and without reference to or discussion of possible PGK1 activity, the differential role of α_1_-AR subtypes, or prior genetic studies. KO of the α_1_-AR subtypes using genetic manipulation and verified through radioligand binding studies have indicated that various aspects of learning and memory are impaired with KO in α_1A_-AR [16], α_1B_-AR [90,91], and α_1D_ [92]. Overexpression of the α_1_-AR subtypes indicates differential roles in the brain, with overexpression of the α_1A_-AR enhancing synaptic plasticity, cognition [12], and lifespan [93], while overexpression of α_1B_-AR leads to neurodegeneration [94,95], synucleinopathy [96], shortened life span [95], and seizures [97].

## 7. Conclusions

Activation of NE-mediated neurotransmission via the AR family (α_1_, α_2_, β) is associated with increased cognitive functions, supported by AR knock-out and transgenic mouse models and clinical trials. However, several prominent studies using nicergoline, which is non-selective, in addition to quinazoline antagonists of the α_1_-AR, which target PGK1 and enhance glycolysis, have suggested that blockage and not activation of the α_1_-ARs, specifically the α_1A_-AR subtype, is a suitable therapeutic pathway to treat neurocognitive diseases. While the cognitive and/or neuroprotective benefits of nicergoline and the “osins” are well-evidenced, their effects are likely due to non-α_1_-AR-mediated activity through “off-target” pathways, and studies using these agents should be interpreted with caution.

In future directions for drug development, while quinazoline α_1_-AR antagonists’ neuroprotective benefits are not α_1_-AR-mediated, this does not decrease their importance as a potential therapeutic for Alzheimer’s disease. The quinazoline scaffolds could be modified to increase affinity for PGK1 and decrease affinity for α_1_-AR, preventing α_1_-AR inhibition. In the development of α_1A_-AR agonists, highly specific agonists would need to be designed, which is possible as the structure of a mildly selective A60613 agonist bound with α_1A_-AR has been published [98]. However, as α_1_-AR agonists increase blood pressure, a drug would need to be signal-biased against the pathways that regulate blood pressure but still activate the cognitive-enhancing pathways. Such is the case for a positive allosteric modulator of the α_1A_-AR that was shown in pre-clinical studies to improve long-term synaptic plasticity and cognition, and clear β-amyloids in AD mouse models better than donepezil (i.e., Aricept) but without effects on blood pressure [34]. With the structure of the α_1A_-AR bound to a mixed allosteric modulator [99], we speculate that the design of additional positive allosteric modulators with greater efficacy and potency may be possible.

## Figures and Tables

**Figure 1 pharmaceuticals-18-01425-f001:**
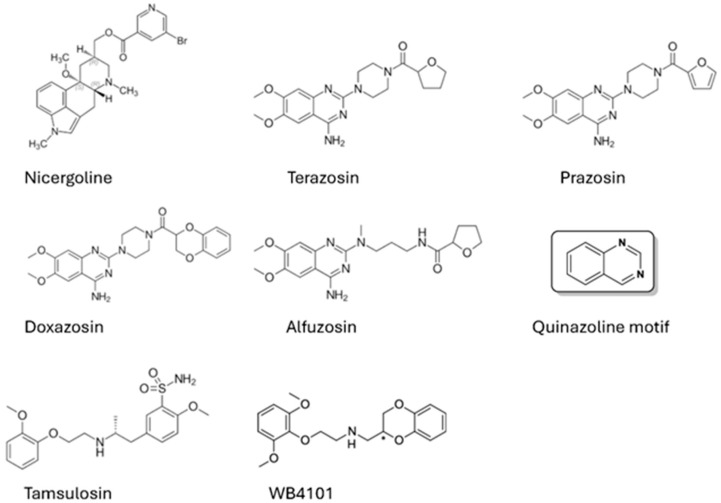
Chemical structures of nicergoline, the quinazoline “osins” (terazosin, prazosin, doxazosin, alfuzosin), and the α_1A_-AR mildly selective non-quinazoline blockers, tamsulosin and WB4101.

## Data Availability

No new data were created or analyzed in this study.

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
