# Peer review of "α1A-Adrenergic Receptor as a Target for Neurocognition: Cautionary Tale from Nicergoline and Quinazoline Non-Selective Blockers"

_pharmaceuticals, 2025, doi:10.3390/ph18101425_

Round 1
Reviewer 1 Report
Comments and Suggestions for Authors
Dianne M. Perez submitted an interesting review about drug discovery based on a1A-adrenergic receptor. The topic was of a certain significance nowadays, and would arouse some discussion in this field. The manuscript could be considered for publication in Pharmaceuticals, but a Major Revision should be conducted first. Please refer to the detailed comments.
- Seem from the length of the manuscript, it should be viewed as a Mini-Review. Please indicate it in the Title.
- The current headings of the manuscript were unacceptable. The Introduction was too long. The reviewer suggested to define the current Section 1.1 as the Introduction, and Section 1.2~1.6 as Section 2~6, correspondingly. Besides, please add a paragraph to showcase the logical framework of this review at the end of the new Introduction.
- In the sections named “NE and AR activation increase memory and cognition” and “a1-AR Activation Increases Cognition and Memory”, the molecular interaction and cellular pathways should be discussed to make it more balanced.
- The pharmacophore analysis regarding Figure 1 should be performed.
- The “osins” drugs should be introduced with names and chemical structures.
- Why did the authors pick tamsulosin for discussion? Was it anything special? Perhaps a discussion on several drugs or analogs will be better.
- Future directions could be implied at the end of the Conclusions.’
- The format of References must be unified.
Author Response
Dianne M. Perez submitted an interesting review about drug discovery based on a1A-adrenergic receptor. The topic was of a certain significance nowadays, and would arouse some discussion in this field. The manuscript could be considered for publication in Pharmaceuticals, but a Major Revision should be conducted first. Please refer to the detailed comments.
- Seem from the length of the manuscript, it should be viewed as a Mini-Review. Please indicate it in the Title.
We received an email from Yalan Ji, assistant editor for Pharmaceuticals, indicating that they would like us to expand the review which is now at 3627 words (previously at 2737) in order to qualify for a regular review. This we have done. Additions are all in red.
2. The current headings of the manuscript were unacceptable. The Introduction was too long. The reviewer suggested to define the current Section 1.1 as the Introduction, and Section 1.2~1.6 as Section 2~6, correspondingly. Besides, please add a paragraph to showcase the logical framework of this review at the end of the new Introduction.
We have now changed the headings according to the reviewer’s suggestions and have added an additional paragraph about the framework of the review at the end of the new introduction
3. In the sections named “NE and AR activation increase memory and cognition” and “a1-AR Activation Increases Cognition and Memory”, the molecular interaction and cellular pathways should be discussed to make it more balanced.
We have now done so. Additional discussions are highlighted in red.
4. The pharmacophore analysis regarding Figure 1 should be performed.
We have added a paragraph discussing the quinazoline pharmacophore.
5. The “osins” drugs should be introduced with names and chemical structures.
We have now modified the text to introduce the names and chemical structures of the osins where first mentioned in the review.
6. Why did the authors pick tamsulosin for discussion? Was it anything special? Perhaps a discussion on several drugs or analogs will be better.
All of the quinazoline α1-AR antagonists are approved drugs to treat benign prostatic hyperplasia (BPH) but also activate PGK1. Tamsulosin is the correct drug for comparison because it is also approved to treat BPH, is an α1-AR antagonist, but does not have the quinazoline motif and does not activate PGK1. In addition, tamsulosin has displayed cognitive enhancing ability, opposite to that of the quinazoline α1-AR antagonists. All of these differences strongly suggests that the quinazoline motif is the crucial determinant and works through a non-α1-AR mechanism.
We have now explained this rationale in more detail in lines 90-110 in red text.
7. Future directions could be implied at the end of the Conclusions.’
We have now added a paragraph on future directions, lines 319-333
8. The format of References must be unified.
We have gone through all of the references for unification
Reviewer 2 Report
Comments and Suggestions for Authors
The manuscript describes in a comprehensive manner the importance of a1A-Adrenergic Receptor as a Target for neurodegenerative processes linked to either protection or impairment and initiation and disease. The global perspective of the cell surface receptors and their role in human physiology gives way to specific Adrenergic Receptors and the currently available drugs to remedy the declining physiological phenotypes. The intertwined biochemical pathways involved in such processes link the presence of a1A ARs with specific molecular drugs through variably operating mechanisms (often misleadingly specific) that act positively and with tolerable specificity yet extensive side effects to the patients. The look at the recent data on AR involvement in alternative pathways of activation, involving nicergoline and quinazoline, offer a new perspective that leads to another window of research opportunities.
The work was carried out competently and deserves due attention. A number of points, however, need to be clarified and/or corrected prior to any consideration. The appropriate comments are remarked upon below:
1.
In the introduction and the first section, the statement “b-ARs couple mainly though Gs to stimulate the enzyme adenylate cyclase to produce cyclic adenosine 3’, 5’,-monophosphate (cAMP), …” contains the term “Gs”. Since this is the first time it appears as an acronym in the text, it should be written out (the entire name). Subsequently, a suitable acronym should be used (not a single letter).
In the follow up paragraph the statement “a2-ARs couple mainly though Gi to inhibit the production of cAMP and are often used to regulate the cAMP levels induced through b-ARs. A well-known regulation of this type is insulin secretion.” contains the term “Gi”. Analogous corrections should apply there.
In the next paragraph, the statement “a1-ARs were the last of the ARs to be cloned and characterized. a1-ARs canonically couple to Gq to activate phospholipase C (PLC) that causes …” contains the term “Gq”. Analogous corrections should apply there as well.
2.
In the same paragraph (lines 73-75), the statement “However, transgenic and knockout (KO) mouse models has identified some key subtype-selective functions that may be targeted for subtype-selective drug development.” should be rewritten to read “However, transgenic and knockout (KO) mouse models have identified some key subtype-selective functions that may be targeted for subtype-selective drug development.”.
3.
In lines 94-98, the statements “The memory/cognitive inducing signals of NE focus around cyclic adenosine monophosphate (cAMP) production, the phosphorylation of cyclic AMP response element-binding protein (CREB), or Exchange Proteins Activated by cAMP (EPAC) [16-18] which a1-ARs can also regulate in addition to the b-AR canonical pathways [19-22] In addition, …” should be corrected to read “The memory/cognitive inducing signals of NE focus around cyclic adenosine monophosphate (cAMP) production, the phosphorylation of cyclic AMP response element-binding protein (CREB), or Exchange Proteins Activated by cAMP (EPAC) [16-18], which a1-ARs can also regulate in addition to the b-AR canonical pathways [19-22]. In addition, …”.
4.
In section 1.4, the statement “While there is substantial evidence that activation of the a1A-AR is the therapeutic route to treat AD, some …” should be corrected to read “While there is substantial evidence that activation of the a1A-AR is the therapeutic route to treat AD, some …”.
5.
In the last sentence of section 1.4, the statement “Considering the non-selective nature of nicergoline, the previous hypothesis that cognition is increased due to blocking a1A-AR activity, does not seem likely.” should be corrected to read “Considering the non-selective nature of nicergoline, the previous hypothesis that cognition increases due to blocking a1A-AR activity, does not seem likely.”.
6.
In lines 222-224, the statement “Tamsulosin also does not appear to mediate anti-inflammatory or neuroprotective effects [59, 71], confirming that quinazolines neuroprotective effects are non-a1-AR mediated.” should be rewritten to read “Tamsulosin also does not appear to mediate anti-inflammatory or neuroprotective effects [59,71], thus confirming that quinazoline neuroprotective effects are non-a1-AR mediated.”.
Following the minor corrections the manuscript could be considered.
Author Response
The manuscript describes in a comprehensive manner the importance of a1A-Adrenergic Receptor as a Target for neurodegenerative processes linked to either protection or impairment and initiation and disease. The global perspective of the cell surface receptors and their role in human physiology gives way to specific Adrenergic Receptors and the currently available drugs to remedy the declining physiological phenotypes. The intertwined biochemical pathways involved in such processes link the presence of a1A ARs with specific molecular drugs through variably operating mechanisms (often misleadingly specific) that act positively and with tolerable specificity yet extensive side effects to the patients. The look at the recent data on AR involvement in alternative pathways of activation, involving nicergoline and quinazoline, offer a new perspective that leads to another window of research opportunities.
The work was carried out competently and deserves due attention. A number of points, however, need to be clarified and/or corrected prior to any consideration. The appropriate comments are remarked upon below:
1. In the introduction and the first section, the statement “b-ARs couple mainly though Gs to stimulate the enzyme adenylate cyclase to produce cyclic adenosine 3’, 5’,-monophosphate (cAMP), …” contains the term “Gs”. Since this is the first time it appears as an acronym in the text, it should be written out (the entire name). Subsequently, a suitable acronym should be used (not a single letter). In the follow up paragraph the statement “a2-ARs couple mainly though Gi to inhibit the production of cAMP and are often used to regulate the cAMP levels induced through b-ARs. A well-known regulation of this type is insulin secretion.” contains the term “Gi”. Analogous corrections should apply there.
In the next paragraph, the statement “a1-ARs were the last of the ARs to be cloned and characterized. a1-ARs canonically couple to Gq to activate phospholipase C (PLC) that causes …” contains the term “Gq”. Analogous corrections should apply there as well.
I have now define Gs in the text as “a GTP-binding protein (G-protein) that stimulates adenylate cyclase”. However, in pharmacology, that is the correct way (acronym) to refer to this class of G-protein as defined by the IUPHAR/BPS guide to pharmacology (www.guidetopharmacology.org) designated by the International Union of Pharmacology (IUPHAR).
We have also define Gi and Gq is a similar manner in the revised text.
2. In the same paragraph (lines 73-75), the statement “However, transgenic and knockout (KO) mouse models has identified some key subtype-selective functions that may be targeted for subtype-selective drug development.” should be rewritten to read “However, transgenic and knockout (KO) mouse models have identified some key subtype-selective functions that may be targeted for subtype-selective drug development.”.
We have now revised.
3. In lines 94-98, the statements “The memory/cognitive inducing signals of NE focus around cyclic adenosine monophosphate (cAMP) production, the phosphorylation of cyclic AMP response element-binding protein (CREB), or Exchange Proteins Activated by cAMP (EPAC) [16-18] which a1-ARs can also regulate in addition to the b-AR canonical pathways [19-22] In addition, …” should be corrected to read “The memory/cognitive inducing signals of NE focus around cyclic adenosine monophosphate (cAMP) production, the phosphorylation of cyclic AMP response element-binding protein (CREB), or Exchange Proteins Activated by cAMP (EPAC) [16-18], which a1-ARs can also regulate in addition to the b-AR canonical pathways [19-22]. In addition, …”.
We have now revised
4. In section 1.4, the statement “While there is substantial evidence that activation of the a1A-AR is the therapeutic route to treat AD, some …” should be corrected to read “While there is substantial evidence that activation of the a1A-AR is the therapeutic route to treat AD, some …”.
We have now revised
5. In the last sentence of section 1.4, the statement “Considering the non-selective nature of nicergoline, the previous hypothesis that cognition is increased due to blocking a1A-AR activity, does not seem likely.” should be corrected to read “Considering the non-selective nature of nicergoline, the previous hypothesis that cognition increases due to blocking a1A-AR activity, does not seem likely.”.
We have now revised
6. In lines 222-224, the statement “Tamsulosin also does not appear to mediate anti-inflammatory or neuroprotective effects [59, 71], confirming that quinazolines neuroprotective effects are non-a1-AR mediated.” should be rewritten to read “Tamsulosin also does not appear to mediate anti-inflammatory or neuroprotective effects [59,71], thus confirming that quinazoline neuroprotective effects are non-a1-AR mediated.”.
We have now revised
Round 2
Reviewer 1 Report
Comments and Suggestions for Authors
Thanks for your revision.